# Morphing Task: The Emotion Recognition Process in Children with Attention Deficit Hyperactivity Disorder and Autism Spectrum Disorder

**DOI:** 10.3390/ijerph182413273

**Published:** 2021-12-16

**Authors:** Cristina Greco, Maria Romani, Anna Berardi, Gloria De Vita, Giovanni Galeoto, Federica Giovannone, Miriam Vigliante, Carla Sogos

**Affiliations:** 1Department of Human Neurosciences, Sapienza University of Rome, 00185 Rome, Italy; cristinagreco.npi@gmail.com (C.G.); maria.romani@uniroma1.it (M.R.); anna.berardi@uniroma1.it (A.B.); gloria.devita@uniroma1.it (G.D.V.); giovanni.galeoto@uniroma1.it (G.G.); federica.giovannone@uniroma1.it (F.G.); miriam.vigliante@uniroma1.it (M.V.); 2Istituto Neurologico Mediterraneo Neuromed IRCCS, 86077 Pozzilli, Italy

**Keywords:** autism spectrum disorder (ASD), attention deficit hyperactivity disorder (ADHD), emotion recognition, facial expression, pediatric

## Abstract

Recognizing a person’s identity is a fundamental social ability; facial expressions, in particular, are extremely important in social cognition. Individuals affected by autism spectrum disorder (ASD) and attention deficit hyperactivity disorder (ADHD) display impairment in the recognition of emotions and, consequently, in recognizing expressions related to emotions, and even their identity. The aim of our study was to compare the performance of participants with ADHD, ASD, and typical development (TD) with regard to both accuracy and speed in the morphing task and to determine whether the use of pictures of digitized cartoon faces could significantly facilitate the process of emotion recognition in ASD patients (particularly for disgust). This study investigated the emotion recognition process through the use of dynamic pictures (human faces vs. cartoon faces) created with the morphing technique in three pediatric populations (7–12 years old): ADHD patients, ASD patients, and an age-matched control sample (TD). The Chi-square test was used to compare response latency and accuracy between the three groups in order to determine if there were statistically significant differences (*p* < 0.05) in the recognition of basic emotions. The results demonstrated a faster response time in neurotypical children compared to ASD and ADHD children, with ADHD participants performing better than ASD participants on the same task. The overall accuracy parameter between the ADHD and ASD groups did not significantly differ.

## 1. Introduction

Faces are complex stimuli that convey social and affective information; recognizing a person’s identity is a fundamental social ability [1]. In fact, people deduce personality traits from the similarity between the morphological features of a person’s face and emotional expressions [2,3]. Individuals affected by autism spectrum disorder (ASD) display impairment in the recognition of emotions and, consequently, also in the recognition of expressions related to emotions, even their identity.

This is one of the reasons that the practical and clinical applications of automatic emotion recognition have been extensively tested and validated in some neurodevelopmental disorders [4,5,6], particularly in participants with ASD and attention deficit hyperactivity disorder (ADHD).

Facial expression recognition involves dynamic and multimodal phenomena. Facial transformation from a neutral expression sends complex signals, which are converted into emotions [7]. The practical and clinical applications of automatic emotion recognition have been extensively tested and validated [8,9,10,11,12].

ASD is known to be associated with difficulties in using facial expressions to convey emotions and deficits in emotional reciprocity [13,14,15,16,17]. Difficulties in understanding emotions through facial expressions affect a person’s ability to appropriately respond to different situations [18,19,20,21,22]. Since the 1970s, impaired emotion recognition has been described in people with ASD. The six basic emotions (happiness, sadness, fear, disgust, anger, and surprise) identified by Ekman have been investigated [23]. However, emotion recognition findings have been inconsistent to date. Some authors have linked ASD to deficits in the recognition of specific subsets of emotions, which have variously included fear, anger, disgust, and sadness [24,25,26,27], while others observed specific deficits in the recognition of anger or surprise [28,29]. Finally, other studies found no evidence of emotion recognition impairments compared to the general population [30]. It appears that the performance of ASD participants on emotion recognition tasks is influenced by different variables, including age. Some authors found that emotion recognition deficits in children aged 5–7 years decreased in late childhood, but no further improvement has been detected during adolescence or adulthood [31]. A meta-analysis carried out by Lozier and colleagues [32] confirmed that ASD was associated with face-emotion recognition deficits, and that the magnitude of these deficits increased with age.

ADHD is a neurodevelopmental disorder characterized by attention deficit, hyperactivity, and impulsivity (for a more in-depth view, see the Diagnostic and Statistical Manual of Mental Disorders-5 (DSM-5)) [14]. A number of studies found that, in addition to impaired executive functions and problematic behavior [33], emotion recognition deficits associated with interpersonal difficulties may be present, similar to those found in ASD [34].

Children with ADHD are less accepted and often rejected by their peers [35,36,37]. These social difficulties are likely to persist into adulthood. Although some theories assume that emotion recognition deficits are explained by general attentional deficits, increasing evidence suggests that they may actually constitute a distinct impairment. Schönenberg et al. [38] found that individuals with ADHD exhibited impaired recognition of sad and fearful facial expressions, while Schwenck et al. [39] did not find significant differences in emotion recognition between children with ADHD and matched controls. Finally, Jusyte et al. [8] found that, compared to controls, children with ADHD exhibited lower accuracy rates across all basic emotional expressions. Although emotion recognition abilities have been investigated in both ADHD and ASD, few studies have directly compared the performance obtained by individuals with the aforementioned disorders. Furthermore, existing studies have measured expression recognition abilities by employing static images of expressions at their highest intensity. For instance, Berggren et al. [40] used the Frankfurt Test for Facial Affect Recognition to compare matched samples of children with ADHD and ASD. They found that the ADHD group responded faster than the ASD group, but they did not find any difference in accuracy between the two groups. The same test was used by Sinzig et al. [41], who reported facial affect recognition deficits in children suffering from ADHD and in children suffering from both ASD and ADHD when compared to healthy controls. Demopoulos et al. [42] compared the social cognitive profiles of children and adolescents with ASD and ADHD using the Diagnostic Assessment of Nonverbal Accuracy-2 (DANVA-2) to measure effects on facial and vocal identification abilities. Both groups performed significantly worse than the normative sample.

The recognition of facial expression involves dynamic and multimodal phenomena. The transformation of the face from a neutral expression sends complex signals, which are converted into emotions [7,8,9,10,11,12,13,14,15,16,17,18,19,20,21,22,23,24,25,26,27,28,29,30,31,32,33,34,35,36,37,38,39,40,41,42,43].

The morphing task has been used since 2001 to evaluate children with ADHD and ASD [44], and it is still used today [45,46,47]. However, studies in the literature have primarily focused on visual updating; studies comparing the two clinical groups (ADHD and ASD) with a control group through human and animated morphing tasks are scarce.

We aimed to investigate the performance of ADHD, ASD, and TD participants with regard to both accuracy and speed in the morphing task and to determine whether the use of dynamic pictures representing cartoon faces could significantly facilitate the process of recognizing emotions in ASD participants.

We hypothesized that children with ADHD would respond faster than ASD participants due to their executive and attention deficits [40], which is in contrast to the findings of Swenck et al. in 2013, who demonstrated no difference in response accuracy or speed between the ADHD and control groups [39]. We also expected that response accuracy would be reduced, particularly for negative emotions [32,33,34,35,36,37,38,39,40,41,42,43,44,45,46,47,48]. Finally, based on the results obtained in 2015 by Brosnan with ASD participants [49], we hypothesized that the use of cartoon faces may facilitate the recognition of emotions as compared to stimuli representing human faces.

Our study aimed to provide a watershed in the understanding of the emotion recognition process. While many studies have investigated this process in ADHD and ASD populations, comparative studies between the two clinical populations are scarce and results have been conflicting, especially when conducted using tasks with dynamic images (representing cartoon faces). Therefore, we used dynamic images of cartoon faces in emotion recognition, with the perspective of using the morphing task as a tool to enhance the expression recognition process at the rehabilitation level.

## 2. Methods

### 2.1. Participants

Sixty-two children in the age range 7–12 years were recruited and assigned to one of three groups: (1) an ADHD combined type (ADHD-C) group; (2) an ASD group; and (3) an age-matched control group.

All children and parents recruited for the first two groups were referred to the Department of Human Neurosciences at Policlinico Umberto I clinic in Rome for a follow-up evaluation, which included recruitment to our study between May and September 2019. The 21 children in the ADHD group and the 20 children in the ASD group were included after meeting the DSM-5 diagnostic criteria for ADHD-C or ASD, respectively [14], and after scoring ≥85 on the Full-Scale Intelligence Quotient (FSIQ). Participants were excluded if their first language was not Italian, in order to avoid bias related to linguistic difficulties, or if they had comorbid medical or psychiatric disorders detected through a review of their medical record, in order to avoid the risk of bias due to the associated diagnosis. The control participants were 21 children with typical development (TD) recruited from a primary school in Rome. The inclusion criteria for this group were an age range of 7–12 years and an average score on the Raven’s Coloured Progressive Matrices (CPM) [50] or Standard Progressive Matrices (SPM) [51] (above the 25th percentile). The exclusion criteria were visual or auditory impairment or neurodevelopmental, neurological, or organic disorders, as determined through an interview with the parents. The study was approved by the ethics committee of Sapienza University of Rome and performed in compliance with the Declaration of Helsinki (2000). Written informed consent was obtained from the participants’ parents.

### 2.2. Morphing Task

All participants were evaluated using a morphing task. The morphing technique is an image processing technique commonly used to metamorphosize from one image to another (e.g., an initial image of a child gradually turns into an image of an adult).

In our study, 24 videoclips with neutral faces gradually developed basic emotions (sadness, anger, surprise, happiness, disgust, and fear). Videoclips were built using FantaMorph (version 5, Abrosoft Co., Eden Prairie, MN, USA), a software used to create morphing images and sophisticated animation effects, including the transformation of the images used.

It should be emphasized that the group of complex emotions (i.e., pride, embarrassment, jealousy) was excluded since these emotions imply the attribution of a cognitive state as well as an emotion and are more dependent on context and culture [52]. They can also be based on belief [53]. Children with TD begin to recognize and verbally label complex emotions, such as embarrassment, pride, and jealousy, at age 7 [54]. Golan et al. [55] suggested that recognition of complex emotions is also impaired in children with ASD.

These videoclips were split into two conditions to compare emotion recognition in human and cartoon faces. These two conditions were chosen on the basis of studies carried out by Jusyte in 2017 [8] and Brosnan et al. in 2015 [49], which analyzed emotion recognition through human animated stimuli (in the first case) and through human and digital animated stimuli (in the second). Our task included a training session and two different conditions (Figure 1 and Figure 2) [56,57].

### 2.3. Training Session

During the training session, participants watched two videoclips in which a neutral expression turned into a sad or happy expression in order to introduce children to the task. All children passed the initial trial and were then admitted to the experiment.

### 2.4. First Condition

In the first condition, we presented greyscale images representing 12 young adults, six females and six males, selected from the Cohn-Kanade database [56,57]. We removed potentially distracting details, such as hair or body parts, in order to focus attention on the faces.

### 2.5. Second Condition

In the second condition, we used colored pictures showing cartoon faces of six females and six males. These images were included in the Facial Expression Research Group Database (FERG-DB) [58]. They were digitally generated using Maya software (version 1, Autodesk, Mill Valley, CA, USA) and the images were created using a 2D renderer (version 7.1.8, Unity, San Francisco, CA, USA) [58].

### 2.6. Procedure

Participants and their parents were informed about the study procedures and all parents provided written informed consent to participate. The administration room, which was the same for the two clinical groups, was located within the school for the control group. The room had a table with a notebook and two chairs (one for the participant and one for the doctor) and no distracting elements such as posters, billboards, or games.

In the training session, participants were instructed by the child neuropsychiatrist to press a button on the keyboard as soon as a face corresponding to the lexical label appeared on the screen.

Each videoclip lasted 7 s and included 60 frames. We recorded answers and response latency on the appropriate answer sheet. In both conditions, videoclips were presented in the same random order obtained by pseudo-randomization for each child. The experiment was conducted using a Microsoft Office PowerPoint presentation (total time about 20 min), shown on a computer with a 10-inch screen and at a distance of 50 cm.

For statistical analysis, SPSS 19 software (International Business Machines Corporation, Armonk, NY, USA) was used. In order to compare the performance obtained by ADHD, ASD, and TD participants with regard to both accuracy and speed on the morphing task, we performed the following analyses:The Chi-square test (χ^2^) was performed to investigate potential relationships between the number of errors (as categorial variables) and the six emotions in the two conditions (human and cartoon faces);Multivariate analysis of variance (MANOVA) was used to investigate group differences with regard to response latency (speed in seconds).

## 3. Results

The clinical participants were 20 children with ADHD combined type (ADHD-C) in the age range 7–12 years old (mean age: 10.12 years; 15 males and 5 females), 21 children with ASD in the age range 7–12 years old (mean age: 10.21 years; 18 males and 3 females), and 21 children in the control group (TD) in the age range 7–12 years old (mean age: 9.33 years; 13 males and 8 females). Demographic characteristics are reported in Table 1.

The Chi-square test was used to compare response accuracy between the three groups in order to determine if there were statistically significant differences (*p* < 0.05; *p* < 0.01) in the recognition of basic emotions. In condition 1 (human faces), ASD children exhibited a significantly higher error frequency as compared to TD participants when they were asked to recognize disgusted faces (0.3% ADHD, 0.4% ASD, 0.2%TD; χ^2^ = 7.612; *p* = 0.022). With regard to surprise, the ADHD group exhibited the highest error frequency compared to the other two groups (0.6% ADHD, 0.2 % ASD, 0.2% TD; χ^2^ = 6.025; *p* = 0.049).

In condition 2 (cartoon faces), we found a significant difference in error frequency distribution relative to the emotion of sadness between the ADHD group and the other two groups (1.0% ADHD, 0.0% ASD, 0.0% TD; χ^2^ = 6.620; *p* = 0.037). With regard to disgust, the ADHD group had the highest percentage of errors (0.4% ADHD, 0.3% ASD, 0.2% TD; χ^2^ = 9.371; *p* = 0.009).

We compared the response accuracy for the aforementioned emotions between the two clinical groups (ADHD and ASD). The results were similar to those obtained in the previous phases of this study. In particular, in condition 1 (human faces) we found no significant differences between the ADHD and ASD groups in identifying the emotion of disgust, while the error frequency distribution for surprise was significantly higher in the ADHD group (0.8% ADHD, 0.2% ASD; χ^2^ = 3.881; *p* = 0.049). In condition 2 (cartoon faces), the percentage of error in identifying the emotion of disgust was higher in the ADHD group relative to the ASD group (0.6% ADHD, 0.4% ASD; χ^2^ = 4.020; *p* = 0.045), and the error frequency for sadness was higher in the ADHD group (1.0% ADHD, 0.0% ASD; χ^2^ = 3.399; *p* = 0.065).

### 3.1. Misidentified Emotions

In condition 1, the emotion disgust was more frequently confused with anger in both clinical groups compared to the control group (0.4% ADHD, 0.4% ASD, 0.2% TD; χ^2^ = 15.899; *p* = 0.045). The ADHD and TD groups tended to confuse fear and the facial expression of sadness more frequently than the ASD group, while the ASD group more often confused fear and surprise compared to the other two groups (sadness: 0.3% ADHD, 0.1% ASD, 0.5% TD; surprise: 0.0% ADHD, 1.0% ASD, 0.0% TD; χ^2^ = 22.579; *p* = 0.032). In condition 2, the ADHD group more frequently confused disgust and anger as compared with the TD group, which confused these emotions less frequently (0.4% ADHD, 0.3% ASD, 0.2% TD; χ^2^ = 13.592; *p* = 0.035).

### 3.2. Response Latency

In order to explore the presence of significant differences between the three groups regarding response latency (parameter speed in seconds), we conducted a multivariate analysis of variance (MANOVA). Wilks’ multivariate Lambda test suggested the existence of statistically relevant differences in response latency between the three groups (λ_Wilks_ = 0.220, F = 1.701, η^2^ = 0.531, *p* = 0.020). To report these results, we report the name of the emotion followed by M (male face) or F (female face) to indicate the gender of the stimulus presented. In the post hoc comparison in condition 1 (Table 2) for the emotion happiness F and disgust M, the TD group responded faster than the ASD and ADHD groups. For the emotions surprise M and F, anger and happiness M, and disgust F, the ASD group responded slower than the ADHD and TD groups. Concerning the emotion fear F, response latency in the ASD group was longer than in the TD group, while the ADHD group responded faster than the ASD group for anger M (Table 2).

In condition 2, the post hoc comparison in Table 3 showed slower response times in the ASD group compared to the other two groups. For the emotions surprise F and sadness M, the ASD group responded slower than only the TD group (Table 3).

## 4. Discussion

In this study, we investigated the behavior (in terms of speed and accuracy) of ADHD and ASD individuals in response to dynamic emotional images in a morphing task. When compared to the ASD and ADHD groups, children with TD performed better in terms of speed, while the ADHD group performed better than the ASD group. In terms of accuracy, there was no significant difference between the ADHD and ASD groups.

The findings in our analysis confirm previous studies that found a deficit in the emotion recognition process in subjects with ASD and ADHD. Our study shows that the poor performance of the ASD group, particularly for the emotion disgust, is consistent with literature data, especially in regard to Lozier’s meta-analysis [32]. Meta-analyses have mainly involved sets of pictures representing faces and have excluded other kinds of tasks (e.g., cartoon faces). In the study by Brosnan et al. [49], a population of teenagers with ASD was tested using dynamic and static stimuli representing both human and digital faces. Digital stimuli facilitated the ASD group in identifying emotions, both dynamic and static, similar to the control sample. The performance of the ASD group in response to the dynamic and static stimuli of human faces remained low compared to the control group. A similar result was shown in our study. We found that digital stimuli facilitated the ASD group in identifying emotions, such as disgust, compared to both the TD and ADHD groups. Our data differ from Berggen et al. [40], who found a quicker response time for identifying emotions in the ADHD vs. ASD group compared to the control group, and no significant difference with respect to the performance of the TD group.

Specifically, in condition 1 (human faces), the emotion disgust was least recognized by the ASD group, while the emotion surprise was least recognized by the ADHD group. In condition 2 (cartoon faces), the highest error frequency was shown for the emotions sadness and disgust, in both cases by the ADHD group vs. the TD and ASD groups.

These results suggest that digital faces/cartoons may facilitate recognition for participants in these groups. The ASD group made fewer mistakes recognizing disgust, one of the hardest emotions to perceive, in condition 2. This facilitation suggests that the ASD group may be able to use atypical strategies to recognize emotions. This hypothesis originated from previous studies [30,31,32,33,34,35,36,37,38,39,40,41,42,43,44,45,46,47,48,49,50,51,52,53,54,55,56,57,58,59] in which “exaggerated” facial expressions were perceived as “typical” expressions in static stimuli animated by participants with ASD [60]. Animated stimuli resulted in an extreme representation of emotion in the details of the face, which leads to easier emotion recognition for these participants. The mistakes in emotion recognition were analyzed. The ASD and ADHD groups confused disgust and fear in human faces, and fear was significantly confused with sadness (TD > ADHD > ASD) and surprise (ASD > other groups). In condition 2, disgust was confused with anger (ADHD > TD). These results confirm the emotional profiles of these two disorders and the tendency, mainly in ADHD subjects, for easy recognition of negative emotions, such as anger. Furthermore, ASD participants better recognized emotions in cartoons.

A qualitative observation showed that in our sample there was a lower error rate in cartoon emotion recognition than in human faces. Happiness is the easiest emotion to recognize. Disgust and fear, especially in condition 1, were the most difficult to recognize. This is consistent with literature stating that fear is easier to perceive than recognize. Of the six basic emotions, fear may be defined as the one that conveys the strongest multisensorial signals. These signals, such as environmental threats, may play an important role in decoding emotions [61].

Disgust, after fear, was the most difficult emotion to identify, and was recognized better in the oldest neurotypical participants (similar to fear). Conversely, in ASD, the recognition deficit increased with age, especially for disgust, fear, and sadness [32]. The variability of results for different emotions reflects differences in emotion recognition development by age [62]. For the speed parameter in the “human faces” condition, a longer response latency of the ASD group was shown for both negative and positive emotions. Furthermore, the ADHD group had a shorter response time for anger M compared to participants with TD and ASD. The cartoon condition showed a significantly longer response latency in the ASD group, as compared to the ADHD and TD groups, in negative emotions such as fear M, sadness M/F, anger F, and disgust F, and in only two positive emotions, happiness M and surprise F. These response latency results confirmed the hypothesis that both clinical groups had difficulty in emotion recognition that seems to be related to inherent social difficulties in communication and atypical interactions.

In brief, the difficulty in complex emotion recognition in these participants was evident by:Major latency in emotion recognition for both clinical groups;Greater emotion recognition error rate compared to the control group;Tendency to confuse some emotions (see fear/sadness, anger/disgust).

We hypothesized that there would be an emotion recognition deficit in children with ASD and ADHD compared to the control TD group when using a dynamic stimulus, such as the morphing test. Overall, a faster response time was detected in the group of children with TD compared to the ASD and ADHD groups. The ADHD group, in turn, had a faster response time than the ASD group. Regarding the accuracy of the overall performance, the ADHD and ASD groups did not show a significant difference, and mostly seemed to be superimposable, while the performance of the TD group was qualitatively better. Our data confirm previous studies in the literature. The advantages of the morphing technique have been proven by evidence. Dynamic stimuli may somehow facilitate the emotion recognition process in both groups because a “dynamic picture” impersonalizes human interactions in a more realistic way. In everyday life, facial expressions are rarely transmitted and decoded as static snapshots of internal states [7]. Dynamic faces represent a richer and more valid view of the way in which emotions and facial expressions are identified [60,61,62,63]. The morphing technique is used in many fields, from healing processes and surgery dynamics [59,60,61,62,63,64] to the emotion recognition process in psychiatric pathologies [65]. Our data also show that the use of digitalized faces/cartoons may represent a means to facilitate recognition in ASD participants.

Considering the remarkable facilitation and improved response to dynamic pictures, we hypothesize that the morphing technique could be a valuable aid to reinforce the bases of mind theory and may support the improvement of social reciprocity in ADHD [66] and ASD participants in a therapeutic context.

Several studies have been performed in order to better understand the utility of computer-based interventions (CBIs) in teaching social and emotional skills to subjects with ASD [67,68].

Ramdoss et al. [69], in a systematic review, concluded that CBIs showed mild effects on social and emotional skills in ASD, and that CBIs may represent a promising practice to improve these abilities. However, the authors underlined several limitations, including the heterogeneity of subjects in terms of age and cognitive capacities. Furthermore, the literature reports have stressed the difficulties that individuals with ASD may have in applying the abilities acquired in CBIs to real-life situations, and has suggested combining this approach with a group activity or face-to-face instruction with an adult tutor in order to improve generalization in ASD individuals.

Consistent with these studies, the findings of the present study evidenced the possibility that the morphing task might be a way to simplify emotion recognition in these populations, allowing them to understand the internal states of others, enhance their empathy, and, hypothetically, reduce their gaps in social relationships. In our study, this improvement occurred above all for disgust. However, it can be hypothesized that an improvement for other emotions could be achieved by perfecting the videos through the contextualization of cartoon faces.

This study has some limitations. First, the clinical sample included children of a specific age (7–12 years old). Extending the sample from preschool to adulthood would allow the emotion recognition process to be evaluated throughout life and would allow the accuracy and response latency to be associated with age in these populations, as Richoz showed for the normotypical population [7].

Second, the visual attention parameter (using eye tracking or configurable face processing skills) was not assessed, and consequently the differences in face perception could represent a bias of the study.

Third, all participants included in the study had a pure diagnosis of ASD or ADHD; none had an overlap of the two diagnoses. In addition, all participants had an FISQ of 85 or more. While there is no evidence that these factors influenced the results obtained, we cannot rule out this possibility.

Fourth, the images used in condition 1 were grayscale images of young adults. This could represent a bias with respect to the use of color cartoon images in condition 2. In fact, it has been suggested in the literature, despite the scarcity of studies, that the use of color images of child and adolescent faces may represent a more realistic confirmation of the ability to recognize emotions in the pediatric population.

Additional aspects of interest may be represented by functional Magnetic Resonance Imaging (MRI) studies and the activation of specific brain structures during the emotion recognition process through dynamic tasks [70] and eye-tracking studies with different setting patterns identified [1]. Finally, we used grayscale images in the static task and colored images in the cartoon task, which may have biased our results. The better performance of our participants may be due to the colorfulness of the stimuli. Future studies should evaluate these variables.

This study demonstrates that emotion recognition was facilitated through the use of cartoon faces in ASD and ADHD populations, and that the quality of emotion recognition was comparable between ASD and ADHD participants. This is the first study to compare facial recognition in participants with ADHD, ASD, and TD using a morphing task.

## 5. Conclusions

The findings of this study may have clinical implications in the planning of alternative strategies for rehabilitation settings for children with neurodevelopmental disabilities. The theoretical implications include a better understanding of the maturational changes underlying social skill deficits. We believe that the morphing task has proven to be useful in behavioral investigations and that its use in a more natural or ecological assessment setting is promising. Finally, these results are important for future research evaluating the emotion recognition process as a therapy goal and the morphing task as a rehabilitation tool in order to improve the social and empathy skills of children with ASD and ADHD.

## Figures and Tables

**Figure 1 ijerph-18-13273-f001:**
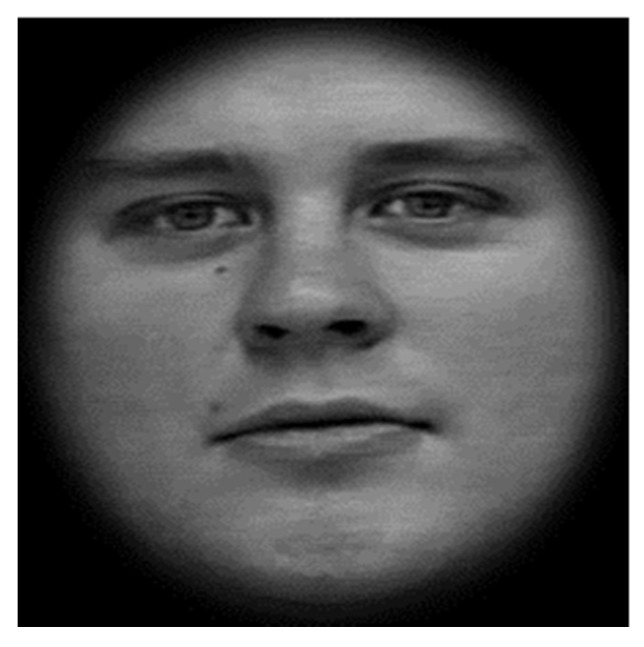
Example of a frame of a human face without distracting elements (adapted from Fantamorph). ©Jeffrey Cohn, S52.

**Figure 2 ijerph-18-13273-f002:**
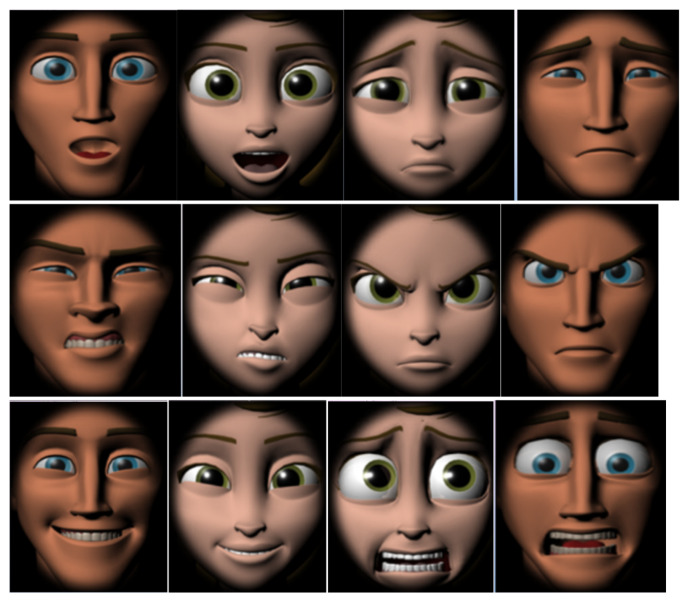
Frames of cartoon faces without distracting elements.

**Table 1 ijerph-18-13273-t001:** Demographic characteristics of the included participants: 21 children with typical development (TD), 20 children with attention deficit hyperactivity disorder (ADHD), and 20 children with autism spectrum disorder (ASD). The clinical participants were 20 children with ADHD.

	ASD	ADHD	TD
Mean age (range)	9.33 (7–12)	10.12 (7–12)	10.12 (712)
Female (number (%))	8 (38)	5 (25)	3 (15)

**Table 2 ijerph-18-13273-t002:** Post hoc comparison: diagnostic groups × latency morphing human faces.

Feeling of Human Face	Groups			Comparison		χ^2^
	ADHD Mean ± SD	ASD Mean ± SD	TD Mean ± SD	ASD vs. TD	ADHD vs. TD	
Happiness_Female Face	4.57 ± 1.764	5.24 ± 1.44	3.24 ± 1.31	*p* < 0.01	*p* < 0.05	0.243
Disgust_Male Face	5.09 ± 1.05	5.23 ± 1.72	4.14 ± 1.55	*p* < 0.05	*p* < 0.05	0.104
Surprise_Male Face	4.73 ± 1.43	5.56 ± 0.86	4.75 ± 1.19	*p* < 0.05	Not significant	0.103
Anger_Female Face	4.71 ± 1.43	5.83 ± 1.13	4.54 ± 1.30	*p* < 0.01	*p* < 0.05	0.173
Fear_Female Face	5.66 ± 1.56	6.62 ± 0.75	5.31 ± 1.43	*p* < 0.01	Not significant	0.161
Happiness_Male Face	4.01 ± 1.06	5.09 ± 1.24	4.22 ± 0.82	*p* < 0.01	*p* < 0.05	0.170
Anger_Male Face	4.86 ± 1.66	5.95 ± 1.07	5.14 ± 1.11	*p* < 0.05	*p* < 0.05	0.115
Surprise_Female Face	3.66 ± 1.01	4.54 ± 1.41	3.61 ± 0.83	*p* < 0.05	Not significant	0.136
Disgust_Female Face	4.39 ± 1.15	5.29 ± 1.44	4.06 ± 1.06	*p* < 0.01	*p* < 0.05	0.159

Note. ADHD = attention deficit hyperactivity disorder; ASD = autism spectrum disorder; TD = typical development; χ^2^ = Chi-square test.

**Table 3 ijerph-18-13273-t003:** Post hoc comparison: diagnostic groups × morphing latency cartoon faces.

Feeling of Cartoon Face	Groups			Comparison		χ^2^
	ADHD Mean ± SD	ASD Mean ± SD	TD Mean ± SD	ASD vs. TD	ADHD vs. TD	
Fear_Male Face	3.56 ± 1.49	4.711 ± 1.823	3.13 ± 1.43	*p* < 0.01	*p* < 0.05	0.157
Sadness_Female Face	3.42 ± 1.09	4.14 ± 1.3	2.97 ± 0.65	*p* < 0.01	*p* < 0.05	0.184
Anger_Female Face	3.04 ± 0.93	4.46 ± 2.17	3.09 ± 1.07	*p* < 0.01	Not significant	0.167
Happiness_Male Face	2.73 ± 0.8	3.71 ± 2	2.35 ± 0.49	*p* < 0.01	*p* < 0.05	0.176
Disgust_Female Face	3.57 ± 1.54	4.7 ± 2.12	3.07 ± 1.24	*p* < 0.01	*p* < 0.05	0.149
Surprise _Female Face	3.46 ± 1.05	4.07 ± 1.91	2.97 ± 0.65	*p* < 0.01	*p* < 0.05	0.111
Sadness_Male Face	3.54 ± 1.4	4.1 ± 1.91	2.8 ± 0.542	*p* < 0.01	*p* < 0.05	0.132

Note ADHD = attention deficit hyperactivity disorder; ASD = autism spectrum disorder; TD = typical development; χ^2^ = Chi-square test.

## Data Availability

The data that support the findings of this study are available from the corresponding author upon a reasonable request.

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
