# Peer review of "Morphing Task: The Emotion Recognition Process in Children with Attention Deficit Hyperactivity Disorder and Autism Spectrum Disorder"

_ijerph, 2021, doi:10.3390/ijerph182413273_

Round 1
Reviewer 1 Report
This manuscript reports the results of a study comparing the performance obtained by ADHD, ASD and TD children in the Morphing task and how the use of digitized cartoons dynamic pictures can facilitate the emotions recognition. The authors discuss these findings in light of prior research and focus on clinical implications.
There are some positive features to this study and manuscript. First, a focus on a variable (emotion recognition ability) having a crucial importance from both clinical and research perspectives. Second, the inclusion of two clinical groups compared to a TD group. Third, objective and well established measures were used as dependent variables. Finally, statistical analyses appeared appropriate for addressing the primary hypothesis of the study.
Despite these strengths, a recommendation for publication is questionable due to some concerns.
Abstract:
It is unusual to start with the purpose of the study without a brief description of the background
Introduction
The introduction for the paper is well documented including issues linked to the focus of the manuscript. The main concern about this study is that the manuscript does not provide a compelling rationale for why this study needs to be conducted relative to the large amount of previous research examining the facial emotion recognition ability in both ASD and ADHD population. In other words, what gap in the literature is this study designed to address? A clear and compelling rationale for the study must be stated, explaining also if it provides a substantive contribution to the extant literature.
Methods section
Lines 116-117: The authors should state why no native Italian participants were excluded.
Moreover, it is surprising that the included subjects had none associated medical or psychiatric condition. This circumstance, in my opinion, weakens the translational power of the study since more than two‐thirds of individuals with ADHD have at least one other coexisting condition and as many as 85% of children with autism also have some form of comorbid psychiatric diagnosis.
Line 128: A distinction between emotion recognition of the six basic emotions (happy, sad, fear, anger, disgust, surprise) and complex emotions (such as jealousy or guilt) should be mention since these last usually involve greater theory of mind and belief-based decision making and since their processing reaches maturity considerably later. It should be stated the kind of emotions studied by the Morphing Task (I suppose only the basic emotions). More in general the ultimate purpose of the task should be better described.
Lines 167-168: I suggest to better clarify the procedure. How the subjects had to identify the corresponding emotion? Did they press a keyboard button as soon as a “written word” describing to the emotion appeared on the screen? Or did they press the keyboard button as soon as a face appropriate to a lexical label appeared on the screen?
Discussion section
Line 342- 343 Finding a way to simplify the recognition of emotions in these populations means allowing them to understand the internal states of others, enhance their empathy and try to reduce their gap in social relationships.
I suggest to the authors more caution in asserting that the improvement in the facial emotions recognition ability ameliorates tout court the understanding of the internal states and, thus, the social competences. Since the study does not provide data on social abilities, it would be better to suggest this as a plausible hypothesis. Furthermore, solid previous researches aimed to demonstrate the direct connection between the facial recognition ability and the social abilities should be cited. For instance, a significant improvement in social behaviour as a result of watching The Transporters (Golan O. et al, 2009; www.thetransporters.com) has been observed in Young RL, Posselt M. Using the transporters DVD as a learning tool for children with Autism Spectrum Disorders (ASD). J Autism Dev Disord. 2012 Jun;42(6):984-91. doi: 10.1007/s10803-011-1328-4. PMID: 21822764). However, many different variables can influence the response to this kind of programs. For instance, limited support for their efficacy is reported for young children with autism of a lower cognitive range (Williams BT, Gray KM, Tonge BJ. Teaching emotion recognition skills to young children with autism: a randomised controlled trial of an emotion training programme. J Child Psychol Psychiatry. 2012 Dec;53(12):1268-76. doi: 10.1111/j.1469-7610.2012.02593.x. Epub 2012 Aug 7. PMID: 22881991.). I suggest to the author to improve the discussion to the light of previous literature data (e.g. Ramdoss S, Machalicek W, Rispoli M, Mulloy A, Lang R, O'Reilly M. Computer-based interventions to improve social and emotional skills in individuals with autism spectrum disorders: a systematic review. Dev Neurorehabil. 2012;15(2):119-35. doi: 10.3109/17518423.2011.651655. PMID: 22494084.). Data on ADHD population is even more spared and rarely takes into account different variables, such as the symptoms’ severity. For instance, a recent paper showed within the ADHD population that emotion recognition accuracy was inversely related to social and emotional problems, but not to prosocial behavior (e.g. Staff, A.I., Luman, M., van der Oord, S. et al. Facial emotion recognition impairment predicts social and emotional problems in children with (subthreshold) ADHD. Eur Child Adolesc Psychiatry (2021). https://doi.org/10.1007/s00787-020-01709-y). The discussion needs to be improved with inferential reasoning on these issues.
Finally, the authors need to be careful of some problems with English expression. A revision by a mother tongue is required.
Author Response
We appreciate the opportunity to resubmit our article entitled “Morphing task: the emotion recognition process in children with attention deficit hyperactivity disorder and autism spectrum disorder.” We would like to thank the reviewers for the careful and constructive review. We have made corresponding changes directly to the manuscript as appropriate, and we have highlighted these changes in the manuscript. The revised version of our manuscript accompanies this letter. All comments by the reviewer have been addressed and are detailed below.
|
The abstract has been modified according to the reviewer’s comment. |
|
The rationale has been enriched with the following paragraph: Our study aimed to provide a watershed in the understanding of the emotion recognition process. While many studies have investigated this process in ADHD and ASD populations, comparative studies between the two clinical populations are scarce and results have been conflicting, especially when conducted using tasks with dynamic images (representing cartoon faces). Therefore, we used dynamic images of cartoon faces in emotion recognition, with the perspective of using the morphing task as a tool to enhance the expression recognition process at the rehabilitation level. |
|
This has been clarified: in order to avoid bias related to linguistic difficulties. |
|
This has been clarified: in order to avoid the risk of bias due to the associated diagnosis. |
|
Given the clinical complexity and age range of the two samples considered, it was not possible to analyze emotion recognition of complex emotions. Line 152 It should be emphasized that the group of complex emotions (i.e., pride, embarrassment, jealousy) was excluded since these emotions imply the attribution of a cognitive state as well as an emotion and are more dependent on context and culture [52]. They can also be based on belief [53]. Children with TD begin to recognize and verbally label complex emotions, such as embarrassment, pride, and jealousy, at age 7 [54]. Golan et al. [55] have suggested that recognition of complex emotions is also impaired in children with ASD. |
|
Children were instructed to press a button on the keyboard as soon as they recognized the emotion/lexical label. |
|
This has been added to the introduction. |
|
The English has been revised. |

Reviewer 2 Report
- Figure 1 is not sufficiently well presenting the test images. The reader cannot build an own opinion how easy/difficult the stimulus data look like. Please give example images (human- and cartoon faces) of all the studied emotion types: fear, sadness female, anger, happiness, disgust, surprise, sadness male.
- The text seems not to be consistent with the entries in table 2 and 3. Please explain what is M (mean?) and sd (std. dev.?) and what they represent – error rates in percentage or latency in msec? What is compared here in the “comparison” column? There seems to be some operator missing here.
- What is the meaning of eta^2? Is it important for any conclusion from the experiments?
- If you observe both the error statistics and the response latency, please provide to types of tables with these two types of results.
- You write (line 194) “determine if there were statistically significant differences (p <.05)”. Comment: I thought if p is greater than 0,05, we believe the variables are independent?
Author Response
We appreciate the opportunity to resubmit our article entitled “Morphing task: the emotion recognition process in children with attention deficit hyperactivity disorder and autism spectrum disorder.” We would like to thank the reviewers for the careful and constructive review. We have made corresponding changes directly to the manuscript as appropriate, and we have highlighted these changes in the manuscript. The revised version of our manuscript accompanies this letter. All comments by the reviewer have been addressed and are detailed below.
|
We do not have the rights to publish the face images, but we have added Figure 2 showing the cartoon faces used. |
|
The captions and tables have been corrected. |
|
The captions have been corrected. |
|
Tables have been corrected |
|
Tables have been corrected |

Round 2
Reviewer 1 Report
The authors have improved the manuscript accordingly to my suggestions
Reviewer 2 Report
Thank you for updating your manuscript according to my remarks.